# Using machine learning for real-time estimates of snow water equivalent in the watersheds of Afghanistan

Edward H. Bair[1], Andre Abreu Calfa[2], Karl Rittger[3], Jeff Dozier[4]

[1]Earth Research Institute, University of California, Santa Barbara, CA 93106-3060, USA
[2]Department of Computer Science, University of California, Santa Barbara, CA 93106-5110, USA, now at Arista Networks, Santa Clara CA 95054, USA
[3]National Snow and Ice Data Center, University of Colorado, Boulder, CO 80309-0449, USA
[4]Bren School of Environmental Science & Management, University of California, Santa Barbara, CA 93106-5131, USA

*Correspondence to*: Edward H. Bair (nbair@eri.ucsb.edu)

**Abstract:**

In many mountains, snowmelt provides most of the runoff. Operational estimates use imagery from optical and passive microwave sensors, but with their limitations. An accurate approach, which we validate in Afghanistan and the Sierra Nevada USA, reconstructs spatially distributed snow water equivalent (SWE) by calculating snowmelt backward from a remotely sensed date of disappearance. However, reconstructed SWE estimates are available only retrospectively; they do not provide a forecast. To
estimate SWE throughout the snowmelt season, we consider physiographic and remotely-sensed information as predictors and reconstructed SWE as the target. The period of analysis matches the AMSR-E radiometer's lifetime from 2003 to 2011, for the months of April through June. The spatial resolution of the predictions is 3.125 km, to match the resolution of a microwave brightness temperature product. Two machine learning techniques—bagged regression trees and feed-forward neural networks—produced similar mean results, with 0–14% bias and 46–48 mm RMSE on average. Nash-Sutcliffe efficiencies averaged 0.68 for
all years. Daily SWE climatology and fractional snow-covered area are the most important predictors. We conclude that the methods can accurately estimate SWE during the snow season in remote mountains, and thereby provide an independent estimate to forecast runoff and validate other methods to assess the snow resource.

## 1 Introduction

Accurate estimates of snow water equivalent (SWE) in mountain watersheds pose a longstanding, unsolved problem. Lettenmaier
et al. (2015) note that "retrieval of snow water equivalent from space remains elusive especially in mountain areas" and argue that "this area deserves more strategic thinking from the hydrology community." Dozier et al. (2016) identify five approaches to the problem, but point out that all are problematic in some way. Operational models' high uncertainty imposes costs for water users. For instance, April to July runoff forecasts in the well instrumented American River Basin in California's Sierra Nevada have a median error of 18% and a 90th percentile error (1 year out of 10) exceeding 60% (Dozier, 2011). Uncertainty stems from the
heterogeneous distribution of mountain snow. A sparse network of sensors sometimes fails to characterize this heterogeneity or even its integrated volume over a drainage basin. In Afghanistan, few stations measure any hydrological or meteorological variables, and only a few dams store snowmelt runoff. Low snowpack years lead to humanitarian crises with little warning, as rivers and streams run dry in the fall and crops fail (e.g. USAID, 2008).

Remotely sensing SWE in the mountains has also proven difficult. Passive microwave (PM) sensors offer real-time global SWE
estimates but suffer from several issues, notably signal loss in wet snow (Li, 2006), saturation in deep snow (Hancock et al., 2013; Kelly et al., 2003; Takala et al., 2011; Tedesco and Narvekar, 2010), decreasing SWE with increasing forest fraction (Nolin, 2010;

Tedesco and Narvekar, 2010), subpixel variability in the mountains owing to the large (~25 km) pixel size (Vander Jagt et al., 2013), and SWE overestimation in the presence of large grains such as depth and surface hoar (Derksen et al., 2005; Durand et al., 2011). A new product, the Calibrated Passive Microwave Daily EASE-Grid 2.0 Brightness Temperature Earth System Data Record (hereafter enhanced resolution PM, Brodzik and Long, 2016) addresses the resolution limitation by providing gridded brightness

temperatures at 3.125 km or coarser spatial resolution by taking advantage of overlapping footprints, at the expense of some added noise. Even given these concerns, PM sensors could in principle provide less biased information if integrated with other datasets. Doing so requires an independent method of estimating spatially distributed SWE.

A strong candidate for such an independent estimate is reconstructed SWE. From satellite-based imagery in the visible through shortwave-infrared bands, we can successfully map fractional snow-covered area ($F_{SCA}$) at sub-pixel resolution (Painter et al.,

2009; Rittger et al., 2013; Rosenthal and Dozier, 1996). From the remotely sensed date of disappearance, snowmelt can be calculated backward to reconstruct SWE for each day back to peak accumulation (Martinec and Rango, 1981). Successful examples of reconstructed SWE include large basins in the Rocky Mountains (Molotch, 2009; Schneider and Molotch, 2016) and the Sierra Nevada (Girotto et al., 2014; Guan et al., 2013; Rittger et al., 2016). When compared with SWE estimates from NASA's Airborne Snow Observatory in the upper Tuolumne River Basin from 2013 through 2015, reconstructed SWE estimates had a 26% Mean

Absolute Error (MAE, Bair et al., 2016) and no bias. In contrast, the Snow Data Assimilation System (SNODAS, Barrett, 2003), the operational model used by the National Weather Service, had a 65% MAE and overestimated snow every year in that analysis. Reconstruction's main advantage lies in its provision of spatially resolved SWE estimates without the need for ground-based observations. Reconstruction's biggest disadvantage is that SWE can only be calculated retroactively after snow disappears, and it works best in areas with clear accumulation and ablation periods. It also cannot assess SWE in the accumulation zones of glaciers,

and the capability of current methods of mapping $F_{SCA}$ on the ablation zones of glaciers has not been assessed (Painter et al., 2012b).

By nature, space-time cubes of retrievals and derived products from satellites are "big data." For this type of information, machine learning has proven particularly effective compared to traditional multivariate statistical techniques. Machine learning techniques such as ensemble regression (Breiman, 2001; Hastie et al., 2009) and neural networks (Hagan et al., 2014) are able to reproduce

nonlinear effects and interactions between variables without assumptions of a functional form. These machine learning techniques are also robust to overfitting in the presence of large datasets.

The goal of this study is to develop and evaluate models using machine learning for SWE prediction in Afghanistan's watersheds. Predictors include static and dynamic variables that are available in near-real time, during the snow season rather than after the snow has melted. The target variable is reconstructed SWE, which we suggest is the closest ground-truth available for data-sparse

regions.

## 2 Study area

Afghanistan's main mountain range, the Hindu Kush (Figure 1), is marked by seasonal drought during the summer and fall. Salang Pass, which straddles the Upper Kabul and Amu Darya watersheds, is the only location in Afghanistan above 3000 m with long term (albeit not current) climate records (Table 1). Snow depth peaks here in April at a maximum of 450 cm and melts out by July.

In comparison, snow depth in Kabul peaks in February at a maximum of 65 cm and melts out by April. The highest peaks in Afghanistan at 7500 m are covered by permanent snow and ice. Historically, about 5% of the country's land area was forested. Decades of war, illegal logging, and a lack of replanting have reduced forest cover to only 2% (United Nations, 2009). The limited

forest cover facilitates remote sensing of the snowpack from optical and microwave instruments, as canopy cover obstructs the view for optical instruments (Raleigh et al., 2013) and interferes with passive microwave emission (Tedesco and Narvekar, 2010). Because of its seasonal summer drought and lack of reservoirs, Afghanistan's water supply is particularly susceptible to year-to-year variations in snowfall.

## 3 Methods

### 3.1 Predictors and target

A mix of static physiographic (Fassnacht et al., 2012) and dynamic variables were used as predictors (Table 2). All variables were computed at or resampled to 3.125 km resolution using Gaussian pyramid reduction or expansion (Burt and Adelson, 1983) for the initial steps and bilinear interpolation for the final step. The 3.125 km resolution was chosen because it is the finest resolution available for the 36 GHz enhanced resolution PM brightness temperature. The study area shown in Figure 1 is the MODIS h23v05 tile, which the snow-covered portions of Afghanistan's watersheds fit into. Elevation dependent variables were calculated from the 30 arc-sec ASTER Digital elevation model.

Only nighttime data from the microwave brightness temperatures were used, to almost always image a frozen snowpack. The time of acquisition for the brightness temperatures varies by up to 30 minutes centered around 01:00 am local Afghanistan time. The 36 GHz V (vertical polarization) brightness temperatures are available at 3.125 km, but the 18 GHz V brightness temperatures are available only at a resolution of 6.25 km, so the 18 GHz brightness temperatures were resampled to the 3.125 km resolution. Likewise, the 10 GHz brightness temperatures are available only at 12.5 km resolution, so they were also resampled to 3.125 km. Two different brightness temperature differences ($T_{18V}$–$T_{36V}$ and $T_{10V}$–$T_{18V}$) were used to account for shallow and deep snow. Use of these three brightness temperature channels in simulations has shown promise for SWE retrievals (Markus et al., 2006), although the $T_{18V}$–$T_{36V}$ is more commonly used (e.g. Kelly, 2009). At the latitude of the Afghanistan, AMSR-E has 2-day repeat coverage, so gaps in brightness temperatures were filled using bilinear interpolation. The brightness temperatures were then smoothed using a 7-day moving median filter. The $F_{SCA}$ and SWE dynamic variables are discussed below.

### 3.2 Reconstruction using the ParBal energy balance model

To compute snow and ice melt, we used the ParBal (Parallel energy balance) model (Bair et al., 2016), a full energy balance snow and ice melt model, run at an hourly timestep and initially at 463 m resolution in a MODIS sinusoidal projection. ParBal, unlike most melt models, does not require total precipitation, the most uncertain term in the water budget for montane areas (Adam et al., 2006; Milly and Dunne, 2002). At any time step $j$, the snow melt $M_j$ is a product of the fractional snow covered area $F_{SCA}$ and the potential melt $M_p$, i.e. the melt if a pixel were 100% snow covered (Molotch and Bales, 2005):

$$M_j = F_{SCA,j} \times M_{p,j} \tag{1}$$

The potential melt is computed using downscaled inputs from reanalysis and remotely sensed data. We briefly summarize this process and refer the reader to Bair et al. (2016) for details. Elevation dependent variables are scaled using the difference between a coarse-resolution digital elevation model (DEM) for CERES (Clouds and the Earth's Radiant Energy System, Rutan et al., 2015) or GLDAS (Global Land Data Assimilation System, Rodell et al., 2004) and a DEM at the 463 m scale. For incoming solar radiation, CERES incoming solar radiation is used to derive the transmissivity, then the local illumination conditions were computed using horizon angles and knowledge of the sun's position. For incoming longwave radiation, a sky view factor (Dozier

and Frew, 1990) is used. Latent and sensible fluxes are computed using exchange coefficients that depend on wind speed. Melt can only occur when the snow surface temperature is at 0°C. Cold content is accounted for to prevent spurious melt when the energy balance is positive, but the snowpack bulk (not surface) temperature is well below the melting temperature. Incoming radiative fluxes and other meteorological variables are from GLDAS and CERES.

### 3.2.1 Inputs to ParBal

Dynamic inputs to ParBal to compute snow and ice melt include $F_{SCA}$, snow albedo, incoming solar radiation, incoming longwave radiation, air temperature, wind speed, and specific humidity. Downscaled forcings from GLDAS at ¼° and CERES-SYN at 1° provided all the energy inputs. By using both GLDAS and CERES, we overcome biases from failures to detect clouds in the shortwave and longwave GLDAS and NLDAS (Cosgrove et al., 2003) products (Bair et al., 2016; Hinkelman et al., 2015; Lapo et al., 2017). Specifically, CERES SYN provided incoming shortwave and longwave fluxes while GLDAS provided air temperature, wind speed, and specific humidity. Note that GLDAS only provides wind speed, not its vector, therefore the terrain-based downscaling approach (Liston and Elder, 2006; Liston et al., 2007) that Bair et al. (2016) used could not be applied. Instead, GLDAS wind speeds were resampled from ¼° to 463 m. Although this resampling leads to errors in wind speeds and is a source of uncertainty, the latent and sensible terms that depend on wind speed tend to be small and usually of opposite sign (Marks and Dozier, 1992). GLDAS and CERES were linearly interpolated from 3 h to 1 h intervals to match the model time step.

Key inputs are the remotely sensed $F_{SCA}$ and snow albedo. We use the products MODIS Snow Covered Area and Grain Size (MODSCAG, Painter et al., 2009) along with the MODIS Dust and Radiative Forcing in Snow (MODDRFS, Painter et al., 2012a). Snow uncontaminated by particulates is rare outside Antarctica (Warren and Wiscombe, 1980), therefore the clean snow albedo estimate from MODSCAG is adjusted for impurities. We then gap fill and smooth these snow property retrievals using a validated algorithm (Dozier et al., 2008; Rittger et al., 2013). The use of a remotely-sensed albedo, to which snowmelt is very sensitive, has been shown to be far more accurate (Bair et al., 2016; Molotch and Bales, 2006) than age-based schemes (Malik et al., 2014; U.S. Army Corps of Engineers, 1956), yet these age-based schemes predominate in snow models (Girotto et al., 2014; Margulis et al., 2015; Markstrom et al., 2015; Molotch, 2009). To compensate for sensitivity of modeled snowmelt to albedo, some SWE reconstruction studies (Guan et al., 2013; Molotch, 2009; Molotch and Margulis, 2008) used ground-based ancillary information to find the occurrence date of snowfall.

Static inputs to ParBal to compute snow and ice melt include: elevations, slopes and azimuths, horizon angles, topographic view factors, and canopy cover. For elevation data, we use ASTER GDEM version 2. Slopes, azimuths, horizon angles, and view factors were then computed from the digital elevation models (Dozier and Frew, 1990). Canopy type and fraction were taken from the Global Land Survey (USGS, 2009).

### 3.2.2 Cold content

To limit early melt, ParBal constrains melt to only occur in a ripe snowpack. In earlier versions, cold content did not need to be accounted for since the model was not run prior to peak SWE. Cold content is not directly modeled since the density and depth of the snowpack are not known. Instead, calculating its proxy uses a scheme modified from Jepsen et al. (2012).

$$M_{p,j} = \max\left(0, \min\left[\sum_{k=0}^{j} Q_{net,k}, Q_{net,j}\right]\right) \tag{2}$$

$Q_{net}$ is the sum of the energy balance terms:

$$Q_{net,j} = R_j + H_j + L_j + G_j \tag{3}$$

$R$ is net radiation, $H$ and $L$ are sensible and latent heat exchanges, and $G$ is conduction, all at time step $j$. In the Jepsen et al. (2012) approach, cold content is reset daily at midnight ($k = 0$ in Eq. 2). Our modification is to reset the cold content daily after sunset, which is a more accurate estimate of when the energy balance becomes negative and daily cold content starts building. Comparisons of melt from ParBal constrained with this cold content scheme and lysimeter measurements at an energy balance site on Mammoth

Mountain, CA (Bair et al., 2015) show excellent agreement (Fig. S1), with both the onset of melt that reached the bottom of the snowpack and seasonal cumulative total melt. Complex routing through the snowpack before reaching the lysimeters likely causes a few discrepancies in the quantities of daily melt (Kattelmann, 2000; Wever et al., 2014).

### 3.3 Validation in California's Sierra Nevada

To better understand the biases and errors in our reconstructed SWE estimates used in Afghanistan, we ran ParBal in the Sierra

Nevada as it was run in Afghanistan. Specifically, as explained in Sect. 3.2.1, meteorological forcings from GLDAS at ¼° spatial resolution and radiometric forcings from CERES at 1° spatial resolution were used. For validation, we used three years of the best spatial SWE estimates available from NASA's Airborne Snow Observatory for the Upper Tuolumne River Basin in the Sierra Nevada. For comparison, the Supplement includes published reconstructions from ParBal forced with National Land Data Assimilation System 2 (Xia et al., 2012) at 1/8° spatial resolution (Bair et al., 2016). $F_{SCA}$ was the same for both GLDAS/CERES

and NLDAS model runs, which Bair et al. (2016) showed to have 15% MAE and 5% bias on average. Cold content was included in the GLDAS/CERES run, but not the NLDAS run.

Fig. S2 and Tables S1 and S2 show full results of the two model runs, which are summarized here. Overall, compared to the NLDAS model run, MAE in the GLDAS/CERES run from peak SWE through melt out decreased from 26% to 22%, but at the expense of some bias which dropped from 0 to –8%. Notably, the GLDAS/CERES model run was slightly more accurate (16%

and 19% MAE for 2014 and 2015 vs. 20% and 31% MAE) than the NLDAS model run in 2014 and 2015, both drought years with SWE depths similar to what we expect in Afghanistan. Also of note is that for the areas with low canopy cover (0 to 0.2), similar to much of the snow-covered areas in Afghanistan, the GLDAS/CERES model run showed a –19% bias, compared to a 12% bias in the NLDAS model run. Full analysis of the biases and model sensitivity is beyond the scope of this study. For the purposes of this study, we conclude that the reconstructed SWE estimates using the GLDAS/CERES forcings show similar accuracy to the

reconstructed SWE using NLDAS forcings.

### 3.4 Validation in Afghanistan

Examination and validation of reconstructed SWE (Figure 2ab) shows absolute and relative ablation curves for all non-glacierized pixels. The interannual variability is high, with a difference of about 2× in peak SWE between the wettest year (2009) and the driest year (2004). Expressed relative to the mean, the peak SWE was 140% in 2009 vs. 70% in 2004. Especially later in the melt

season, some years show values in excess of 200% of the mean while others show values less than 25% of the mean (Figure 2b).

To our knowledge, there are too few in situ measurements of SWE in Afghanistan during the 2003 to 2011 study period to compare with reconstructed SWE. Instead, we used passive microwave-based SWE estimates from AMSR-E from a previous study (Daly et al., 2012); Table 3 shows comparisons of SWE on April 1st.

The Amu Darya and the Kunar basins were excluded because of extensive glaciers, such that passive microwave estimates of SWE

do not provide meaningful estimates of snow on the ground. In every basin on April 1[st], there were many areas with more

reconstructed SWE than the passive microwave saturation limit of 150 mm (Hancock et al., 2013; Kelly et al., 2003; Takala et al., 2011; Tedesco and Narvekar, 2010). Basin-wide, differences between the results of this study and the results reported in Daly et al. (2012) were small, < 8 mm, for all basins except the Upper Kabul (22 mm difference), which had the highest average SWE in both studies.

At North Salang or Salang Pass, the only high elevation station with long-term climatological measurements available in Afghanistan (Table 1), the Oct-May precipitation is 961 mm. October to May corresponds to the time when most of the precipitation falls as snow, based on the mean temperature. Using an energy balance model, Schulz and de Jong (2004) estimate that on average 44% of the snowpack sublimated in the climatologically similar Atlas Mountains of Morocco. Likewise, Bair et al. (2015) also estimate that on average 44% of the snowpack sublimated at a high altitude site in California's Sierra Nevada using

co-incident lysimeters and a snow pillow. Using this sublimation estimate leaves 538 mm of SWE at the peak, assuming negligible mass gain from condensation/deposition. The mean April 1st reconstructed SWE, which can be assumed to be close to the peak, for Salang Pass is 498 mm.

Since 1983, when weather stations in Afghanistan stopped submitting measurements to the World Meteorological Organization (e.g. Table 1), there have been no in situ snow measurements accessible within the country, at least to the authors of this study. In

2015, this changed when FOCUS Humanitarian Assistance, an affiliate of the Aga Khan Development Network, established 82 Weather Monitoring Posts, hereafter called stations, in Afghanistan, Pakistan, and Tajikistan (Chabot and Kaba, 2016). These stations were established for aid in operational avalanche forecasting as a response to widespread and deadly avalanche cycles that occur often in these countries, but were especially catastrophic in 2012 and 2015. In the fall of 2016, we submitted a request to FOCUS Humanitarian for access to a central database of measurements from these stations. Our request was granted a year later.

Digital records only go back to the fall of 2016, but are available for 96 stations, and 88 stations recorded measurable snow on the ground (Figure 3). The snow measurements are all manual, and include new (24-hr) snow, and total snow depth. Maximum and minimum daily temperature, wind speed/direction, and 24-hr manual rainfall were also recorded.

Although measurements from the stations do not cover our 2003-2011 study period, they are a unique validation source in a region where snow measurements are otherwise nonexistent. Therefore, we ran reconstructed the WY 2017 snowpack with ParBal for the

h23v05 tile and validated our SWE estimates using the station snow measurements.

For quality control, we applied an outlier filter (Hampel, 1974) to the time series of snow depth for each station. Time series were then plotted for each station and spurious values (particularly zeros surrounded by high values that were not caught by the automated filter) were manually removed. Of the 96 stations, 15 had snow depth time series that were too erratic or missing too many values for scientific use. Measurements from these 15 stations were discarded.

For the remaining 81 stations, the snow depth estimates were transformed to SWE using a snow climate classification (Sturm et al., 1995) and associated density model (Sturm et al., 2010). For each station, the Nov-Mar daily mean temperature was used to calculate cooling degree months (CDM). The 24-hr new snow and rainfall observations were used to estimate the daily precipitation $P$, with the simplified assumption of $100\,\text{kg m}^{-3}$ density new snow. Wind speed at the time of observation was used as a daily wind speed since 24-hr average wind speeds were not available. The CDM were combined with Nov-Mar mean daily precipitation ($\bar{P}$)

and mean daily wind speed ($\overline{WS}$) and feed through a classification tree (Figure 8 in Sturm et al., 1995) to determine the snow climate. Boundary values used were: 125ºC for CDM; 2 mm/day $\bar{P}$; and 1.25 m/sec for $\overline{WS}$.

Even though some of these stations were over 3000 m in elevation, they tended to have relatively warm (around melting) Nov-Mar mean air temperatures, so they all classified in the warm snow types: alpine, maritime, and prairie (Figure 4).

Of those three classifications, alpine was predominant (N=49), characterized by low precipitation and low wind speed. Some of the stations with heavier precipitation classified as maritime (N=14) which is a bit misleading given that they are several thousand km from an ocean or sea. Prairie was the least common (N=11), classified by warm temperatures, low precipitation, and high winds. A few of the stations (N=7) were missing more than 10 observation dates for a given month from Nov-Mar, so they were not classified. Instead, we set those stations to alpine, the modal value. Also, after classification, we noticed that some of the prairie stations had high peak snow depths (> 50 cm). We suspect that the new snow was not being consistently or accurately recorded for these stations, so we set those prairie stations to the modal snow class (alpine).

With the snow climate determined, a snow density model (Eq. 6 in Sturm et al., 2010) was used to estimate the bulk density of the snowpack. The density model coefficients (Table 4 in Sturm et al., 2010) vary with snow climate. Expected relative SWE bias from this model based on validation from alpine and maritime areas is -12% to +26% of SWE (Sturm et al., 2010).

The bulk density was used to estimate SWE for each station. Reconstructed SWE values (modeled) were then compared with this SWE estimate from snow depth (measured). For comparison, we assume the MOD09GA gridded product, from which our $F_{SCA}$ estimates are based, has a geolocational uncertainty of 0.5-1.0 pixel (250-500 m), depending on the sensor angle (Tan et al., 2006; Xiaoxiong et al., 2005). Thus, we examined the closest pixel to each station, as well as the 8 neighbors surrounding that pixel. For each of those 9 pixels, we further assumed that the SWE at each station could vary from 88% to 126% of the value calculated with the density model. We refer to these variations in location and density as uncertainty. Taking into account the geolocational and density model uncertainties, we chose the reconstructed SWE that most closely matched the computed value at each station, adjusted each for both sources of uncertainty.

For three selected dates, the agreement between modeled and measured values is excellent (Figure 5). These dates were selected to approximate peak SWE for most of the stations. Mean absolute error (MAE) is 7-18 mm and the bias, relative to the measured mean value, is -3% to +4%. For all stations for all dates (Mar to May), modeled results are unbiased with an MAE of 5 mm (13%).

We also examined the time series from Mar to May for 9 stations with deeper snowpacks (> 100 mm SWE on Mar 1). Plotted are stations with negative bias (Figure 6a-c), no bias (Figure 6d-f), and high bias (Figure 6g-i). Maximum negative bias is -113 mm (-12%) for Ruah (Figure 6a). Maximum positive bias is 76 mm (26%) for Arakht (Figure 6i).

Picking the best match from a 9-pixel neighborhood with a -12 to 26% adjustment for density uncertainty is a best-case scenario. To bound our errors, we also matched our modeled SWE values using three other scenarios that show higher errors: a) 9-pixel neighborhood, but with no adjustment for density uncertainty; b) an adjustment for density uncertainty but no use of a neighborhood; and c) no use of a neighborhood and no density adjustment.

Examination of all 4 scenarios shows that the uncertainty in density is the largest source of error, as the inclusion of a neighborhood with uncertainty in density barely improves the error statistics (Table 4). In relative terms, the MAE ranges from 14% to 102% and the bias ranges from -53% to +41% when one of more sources of uncertainty are not accounted for. These uncertainties are inherent when using snow depth point measurements to verify SWE estimates for 0.5 km pixels. The finding that uncertainty in density accounts for most of the error only reaffirms the need for in situ SWE measurements in the region. Nonetheless, these FOCUS measurements are the only in situ snow measurements accessible to us and likely available to anyone. Especially with the sources of uncertainty accounted for, the agreement of these in situ measurements with our reconstructed SWE values and the lack of bias in our modeled values are encouraging.

## 3.5 Machine learning modeling

Given the size of the dataset, 104 million predictor and target observations, random subsampling was used to keep the computation times reasonable. $F_{SCA}$ values of zero were excluded since $(F_{SCA} = 0) \stackrel{\text{def}}{=} (SWE = 0)$. Including predictions of zero SWE would dramatically lower our error, but would imply false accuracy. At 3.125 km, about $1/3^{rd}$ of the pixels in the study area showed some
snow or ice on every day of the year (Figure 1, blue area). Since reconstruction cannot provide a meaningful estimate of SWE on a glacier, we also excluded those pixels from the analysis. Most of these pixels are in the Amu Darya River Basin, but some are in the Kunar Basin. April through June observations were used since reconstructed SWE estimates are valid only during ablation. Probably, some snow-covered pixels were still in the accumulation phase in early April, but most were likely melting during the day, based on the limited snow climate observations for Afghanistan. The study period could have been extended further into the
summer, but was stopped at the end of June to keep the dataset size manageable. Note that that the SWE reconstructions were computed over the entire 2003–2011 study period, since the SWE is built up from melt-out backwards, but only April through June SWE estimates were used for training.

To better understand the relationships between variables, we first computed a correlation ($r$) matrix between the predictors and the target for 90,000 predictor & target pairs across all years (Figure 7). Between predictors, the *Latitude/Longitude* with *SW/W*
*distance to ocean*; the *SW/W/NW barrier difference*; and the *Mean Reconstructed SWE* with $F_{SCA}$ all had $|r| > 0.70$, indicating strong correlation. For the target *Reconstructed SWE*, only *Mean Reconstructed SWE* and $F_{SCA}$ had $r > 0.70$. We note that $F_{SCA}$ was used as a both a predictor and to compute the reconstructed SWE, the target. Likewise, the daily *Mean Reconstructed SWE* was computed across all years, excluding the year currently being predicted, and used as a predictor. Correlation between predictors and targets is desirable but correlation between predictors, called collinearity, is not. Collinearity does not degrade the performance
of the models, but makes assessment of the importance of the correlated predictors independently more difficult (Dormann et al., 2013), an issue we address in the results.

A half-dozen different approaches were tried on randomly selected subsets of data from all years: stepwise multiple linear regression; support vector machines; cross-validated regression trees; least-squares (LS) boosted regression trees; bagged regression trees; and feed-forward neural networks. The bagged trees and feed-forward neural networks were selected for
consistently producing the lowest root mean squared error (RMSE).

### 3.5.1  Bagged trees

Classification and regression trees (CART) comprise a supervised machine learning technique where predictors are recursively split to produce increasingly homogenous subsets of target data (Breiman et al., 1984). Classification trees are used for discrete, categorical data while regression trees are used for continuous data. This technique has been used extensively on snow-related
prediction, for example in snow mapping (Rosenthal and Dozier, 1996) and avalanche forecasting (Davis et al., 1999; Peitzsch et al., 2012), but it tends to overfit. Bagged (bootstrap-aggregated, Breiman, 1996) trees, with the most recent developments called random forests (Breiman, 2001), are ensemble methods whereby multiple trees are grown from random subsets of predictors, producing a weighted ensemble of trees. Bagged trees reduce overfitting and increase predictive accuracy over single trees (Breiman, 2001). Regression ensemble hyperparameters were optimized by minimizing cross-validation loss; the parameters
included the method (i.e. bagged or LS boosted), number of learning cycles, maximum number of splits, minimum leaf size, and number of variables to randomly sample.

### 3.5.2    Neural networks

Neural networks attempt to mimic biological systems by creating a network of interconnected and weighted elements called neurons (Hagan et al., 2014). The neurons provide the connection between the predictors and the target. In supervised learning, the weights of the neurons are iteratively adjusted to minimize error. For our study, a feed-forward neural network was selected for simplicity. One set of parameters to optimize, the number of neurons in each hidden layer, was chosen by testing different combinations on the randomly selected subsets.

### 3.5.3    Training and prediction

For the final model training, each year from 2003 through 2011 was run separately. For each year, 80,000 randomly selected predictor and target observations were selected for training from all years except the current target year. The trained models were then used to predict 10,000 targets for the target year. Larger sample sizes of 800,000 training observations were tried but did not improve model accuracy. For each basin, 90,000 randomly selected predictor and target observations were used with 20% of the observations held out for validation. Residuals and summary error information, RMSE and bias, were stored for each year. All years were then run by basin to examine regional differences in model performance.

Bagged trees and neural networks have been referred to informally as *black boxes*, because the model structure is complicated and difficult or impossible to interpret, but regression trees offer the ability to independently assess predictor performance. For a single tree, estimates of predictor importance are computing by summing changes in the model mean squared error using every split for a predictor, then dividing by the total number of splits where that predictor appears (Breiman et al., 1984). This method can be repeated over all the trees in the ensemble and then averaged to obtain an ensemble predictor importance (Friedman and Meulman, 2003). We used this technique to examine predictor importance, which ranges from 0 to 1, for the bagged trees. We also tried the out-of-bag permuted predictor importance (Breiman, 2001), but found that it showed high importance for predictors not well correlated with the target. One possible explanation is that correlated predictors can be overestimated using this method (Strobl et al., 2008). Thus, we only used the predictor importance as Friedman and Meulman (2003) define.

## 4  Results and discussion

The bagged trees and the neural networks performed similarly when grouped by year (Figure 8ab). Mean bias across all years is 14% for both methods, and RMSE values are 46 to 48 mm. Year-to-year error is similar, with errors ranging from 29 to 86 mm RMSE with –49 to +94% bias for the bagged trees and 31 to 87 mm RMSE with –48 to +88% bias for the neural networks. Error plots were almost identical, so only those for the slightly better performing bagged trees are shown (Figure 8b).

The worst misses came in 2004, where both models showed high bias, and 2009, where both models were low. Those years coincide with anomalously low and high total SWE volumes, especially at the beginning of the melt season (Figure 2). Similarly, when grouped by basin (Figure 9ab) the bagged trees and neural networks had nearly identical performance with mean values across all basins of 22–28 mm RMSE and 0% bias. The RMSE ranged from 6 to 45 mm RMSE for the bagged trees and 9 to 58 mm for the neural networks. Given that the bias only ranged from 0–1% for the bagged trees and was 0% for the neural networks, it is clear that the higher RMSE is the result of different mean SWE in the basins. The highest RMSE occurred in the Upper Kabul for both models, which showed a much greater mean April 1st SWE value compared to the other basins (Table 3).

Both models showed 0% bias for this basin, indicating excellent performance relative to the other basins. Again, error plots were almost identical, so only those for the bagged trees are shown for brevity (Figure 9b).

To put these results in context, we examine performance in terms of the Nash-Sutcliffe (NS) efficiency (Nash and Sutcliffe, 1970), where the reconstructed SWE is treated as the observed result and the bagged tree/neural network SWE prediction is treated as the forecast. The NS efficiency for all years is 0.68 and ranges from 0.21 (in 2004) to 0.90 (in 2006). All years are greater than zero, indicating more skill than a mean forecast. By basin, NS efficiencies are higher averaging 0.89 and ranging from 0.87-0.91,

showing that most of the forecasting challenge occurs year-to-year. Given the lack of water resources monitoring and management in Afghanistan and the high proportion of runoff from snowmelt (Daly et al., 2012; Vuyovich and Jacobs, 2011), our machine learning predictions could improve runoff predictions, which are likely based on mean conditions.

Figure 10 shows the predictor performance, ranked by importance, a relative measure ranging from 0 to 1 explained in Sect. 3.5.3. As expected, $F_{SCA}$ and *Mean Reconstructed SWE* are by far most important predictors, with values of 0.47 and 0.26. As mentioned

in Sect. 3.5, these two variables are highly correlated ($r = 0.73$) making collinearity a concern. For predictive purposes, collinearity is not an issue, but for predictor importance the collinearity means that it is not possible to separately evaluate the predictive power of $F_{SCA}$ and *Mean Reconstructed SWE*. Clearly both are important predictive variables, but $F_{SCA}$ varies from year to year and therefore would better capture interannual variability.

All other predictors show importance < 0.08. The third most important predictor is *Elevation*, shown to be an important predictor

in previous studies (Fassnacht et al., 2003; Fassnacht et al., 2012; Schneider and Molotch, 2016). The fourth most important variable is *Longitude*, followed by $TB_{18V} – TB_{36V}$, the difference between microwave brightness temperatures at 18 and 36 GHz, showing that the passive microwave SWE retrievals have little predictive power.

Afghanistan is ideal for optical remote sensing, given the sparse canopy. Cloud cover is a serious limitation to optical remote sensing, but the melt season in Afghanistan is usually clear. Previous work has shown that the viewable gap fraction adjustment

used here produces accurate results for under canopy snow mapping (Raleigh et al., 2013). Likewise, the lack of canopy cover in Afghanistan makes it an ideal study area for passive microwave remote sensing, but the results here are not encouraging. A detailed time series of $TB_{18V} – TB_{36V}$ and reconstructed SWE for the pixel containing Salang Pass (Fig. S3) shows the $TB_{18V} – TB_{36V}$ time series to have little year to year variability, limiting its utility for discerning wet years from dry years. The little predictive power for estimating SWE from passive microwave, even at the enhanced 3.125 km resolution in this ideal study area, is discouraging,

although we note that the basin wide SWE volumes estimated by reconstruction were close to those estimated via passive microwave (Daly et al., 2012). However, the snowpack was thin and much of the area in each basin was free of snow on April 1st, suggesting that the passive microwave may only be effective at mapping the presence or absence of snow rather than its depth. Given that passive microwaves are not affected by cloud cover, those data could be used to fill gaps caused by cloud cover in estimates of $F_{SCA}$ derived from optical instruments.

As discussed in Sect. 3.3, our reconstructed SWE estimates are not perfect. Potential sources of uncertainty include unobservable $F_{SCA}$ during cloudy periods, reflectance bias at higher sensor zenith angles (Dozier et al., 2008), melt out date (Slater et al., 2013), snow albedo, and wind speed. We assert however that all of these sources of error have been vetted and addressed in depth (e.g. Bair et al., 2016; Dozier et al., 2016; Molotch et al., 2010; Rittger et al., 2016). The comparison with in situ measurements from the region are unbiased and have very low error when uncertainty is accounted for (Section 3.4).

These models could be implemented operationally assuming that the dynamic inputs are made available in near-real time, certainly feasible for the enhanced resolution PM brightness temperatures and for a gap-filled and smoothed MODSCAG $F_{SCA}$. Given the low predictor importance of the enhanced resolution PM brightness temperatures, that predictor could be excluded or substituted with a PM SWE product that is operationally available (e.g. AMSR2 SWE at 10 km resolution, Kelly, 2013). Likewise, the

MODSCAG F$_{SCA}$ is available now in near-real time from the NASA JPL Snow Data System, although it suffers from noise, gaps, and cloud-snow discrimination issues (Dozier et al., 2008). Another issue to overcome would be application of the models outside the training period, i.e. the accumulation period.

## 5 Conclusion

We successfully constructed two machine learning models, trained on reconstructed SWE, that could be used for operational SWE prediction in the austere watersheds of Afghanistan. We hypothesize that our reconstructed SWE estimates are the most accurate ground truth available for this region. Using novel in situ snow measurements from Afghanistan's watersheds, we show our model to be unbiased with low errors when measurement uncertainty is accounted for. Many areas in Afghanistan show an order of magnitude more SWE than passive microwave sensors indicate. The bagged regression trees performed slightly better than the neural networks and have the added benefit of predictor importance estimates. On average over all years, the models were unbiased, but over-estimated SWE in the lowest snow year and under-estimated SWE in the highest snow year. Moreover, the RMSE was higher in the basins with deeper snow, most notably the Upper Kabul, but unbiased across all basins, indicating consistent performance. Nash Sutcliffe efficiencies average 0.68, with all years greater than zero, indicating improved skill over a mean forecast. The most important predictors were the *fractional snow-covered area* and the *Mean reconstructed SWE* (excluding the year being predicted). *Elevation*, *Longitude*, and a measure of SWE from the passive microwave were the third through fifth most important predictors. The method can provide seasonal snowmelt runoff forecasts based on satellite data alone, based largely on fractional snow-cover and reconstructed SWE estimates.

## 6 Data availability

Most of the data used are in government archives: Calibrated passive microwave daily brightness temperatures are from the National Snow and Ice Data Center [https://nsidc.org/pmesdr/]. MODSCAG and MODDRFS snow cover and dust radiative forcing data are from the NASA JPL Snow Data System [https://snow.jpl.nasa.gov/portal/]. Land survey data are from the Global Land Cover Facility [http://glcfapp.glcf.umd.edu/data/gls/]. ASTER-derived elevation grids are from the ASTER Global Digital Elevation Model [http://dx.doi.org/10.5067/ASTER/ASTGTM.002]. The sparse climate data available collected by the World Meteorological Organization are available from the U.S. National Climatic Data Center [ftp://ftp.atdd.noaa.gov/pub/GCOS/WMO-Normals/RA-II/AH]. GLDAS data are from NASA GSFC's HDISC: Hydrology Data and Information Services Center [https://disc.sci.gsfc.nasa.gov/hydrology]. CERES SYN data at 1°, 3 h resolution can be ordered at the CERES HDF Data Products page [https://ceres.larc.nasa.gov/compare_products-ed2.php]. The reconstructed SWE values used as our training target dataset are available at ftp://ftp.snow.ucsb.edu/pub/org/snow/products/reconstruction/h23v05.

## 7 Supplement link

(to be provided by Copernicus)

## 8 Author contribution

Mr. Calfa originally began the study of machine learning for SWE reconstruction as part of his master's thesis, co-supervised by Prof. Dozier. Dr. Bair and Prof. Dozier subsequently carried out most of the analyses, supported by Dr. Rittger's work on SWE

reconstruction using MODIS and NLDAS data. Dr. Bair produced the first draft of the manuscript, which was subsequently edited by Prof. Dozier.

# 9 Competing interests

The authors declare that they have no conflict of interest.

# 10    Acknowledgements

This work was supported by NASA awards NNX12AJ87G and NNX15AT01G, U.S. Army Cold Regions Research and Engineering Laboratory award W913E5-16-C-0013, and an award from Microsoft Research for computing and storage on Microsoft Azure. We thank Jessica Lundquist, Steven Fassnacht, and one anonymous reviewer for their constructive comments.

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

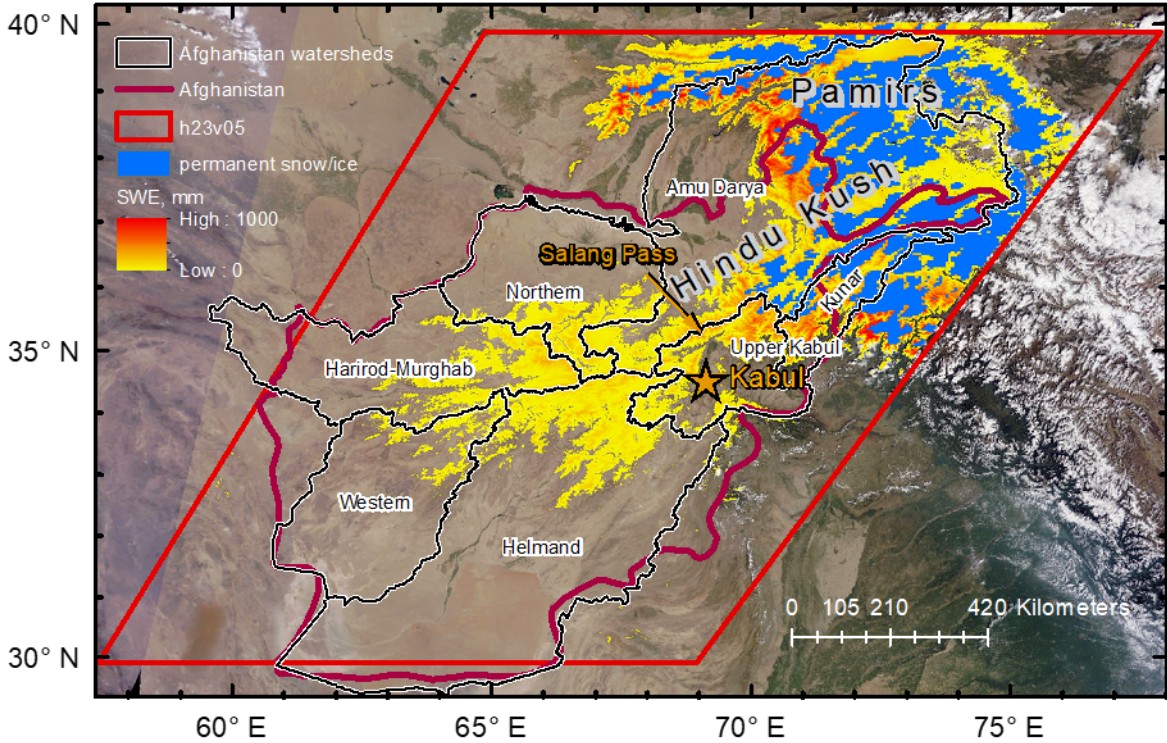

**Figure 1. MODIS true color image of snow covered watersheds of Afghanistan with 2003–2011 April 1st reconstructed mean SWE overlaid. Also shown are the watersheds of Afghanistan [from Daly et al., 2012], the country outline, the h23v05 MODIS tile, and areas with permanent snow and ice.**

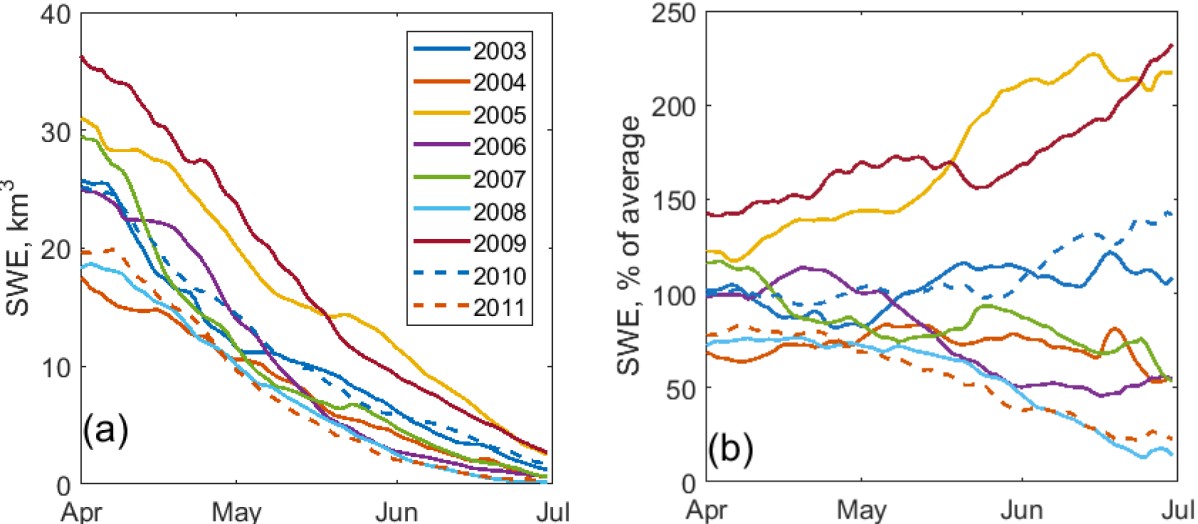

**Figure 2. Reconstructed total SWE volume (a) and percent of 2003–2011 average (b) for all non-glacierized pixels in the study area.**

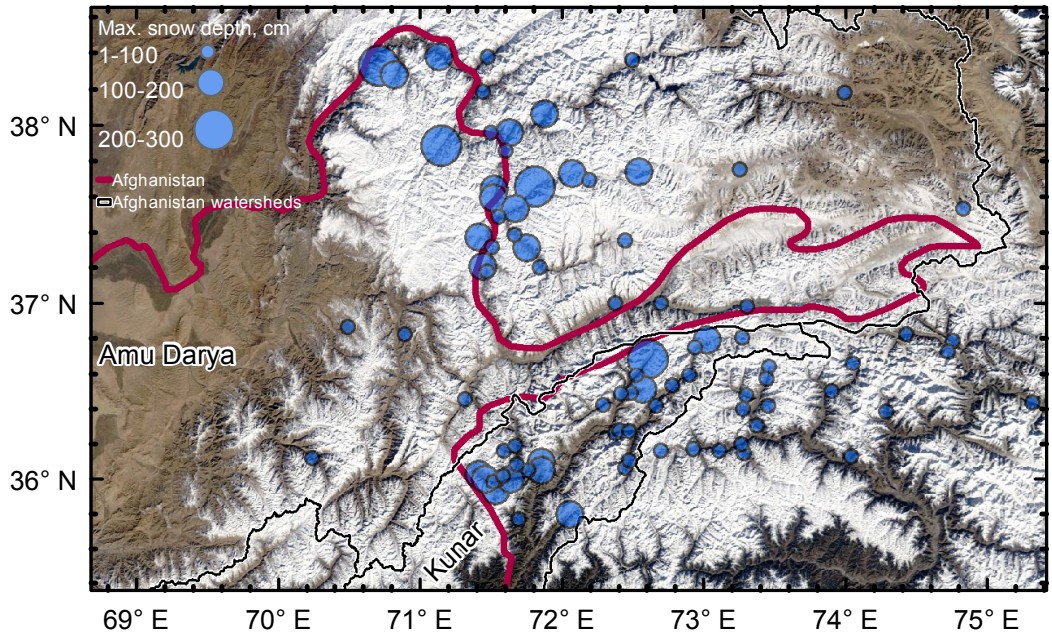

**Figure 3. Maximum snow depth from WY 2017 manual measurements at FOCUS stations in Afghanistan, Pakistan, and Tajikistan overlaid on a MODIS true color image.**

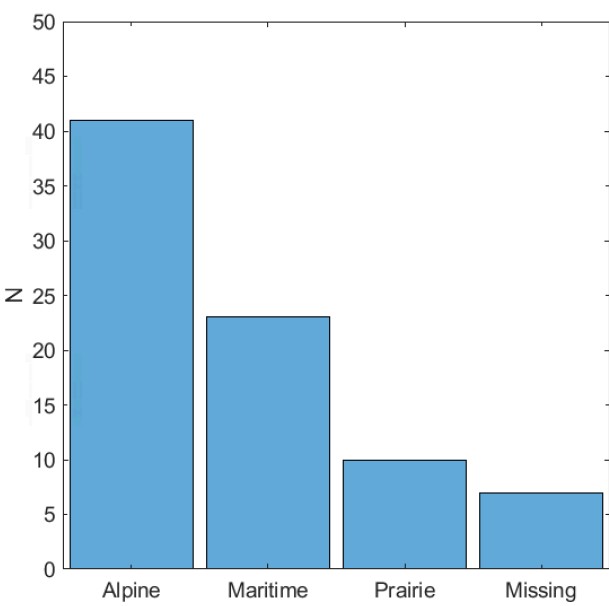

5    **Figure 4. Snow climate classes for 81 FOCUS stations.**

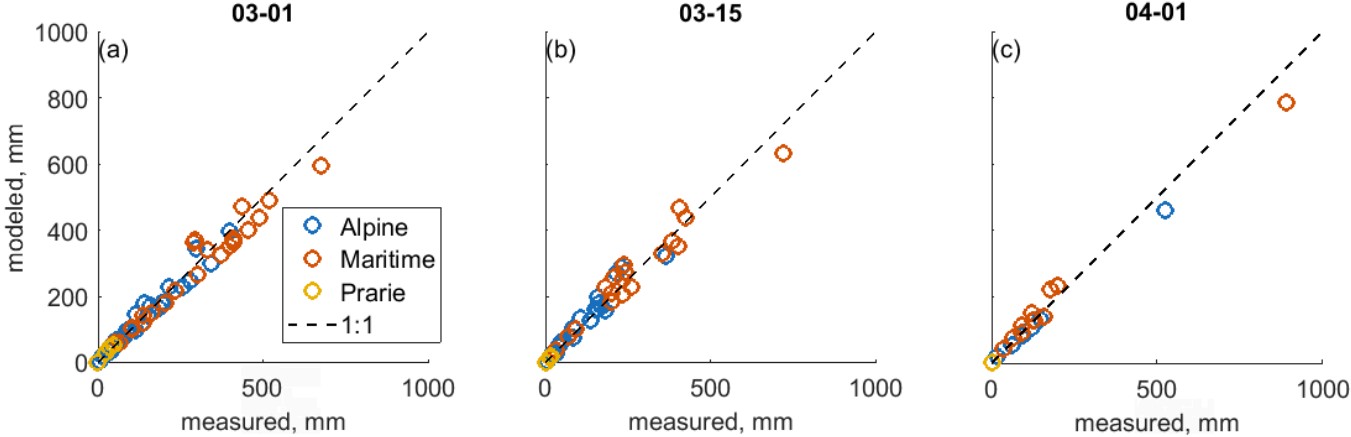

**Figure 5. Model vs. measured SWE at all stations for selected dates in 2017.**

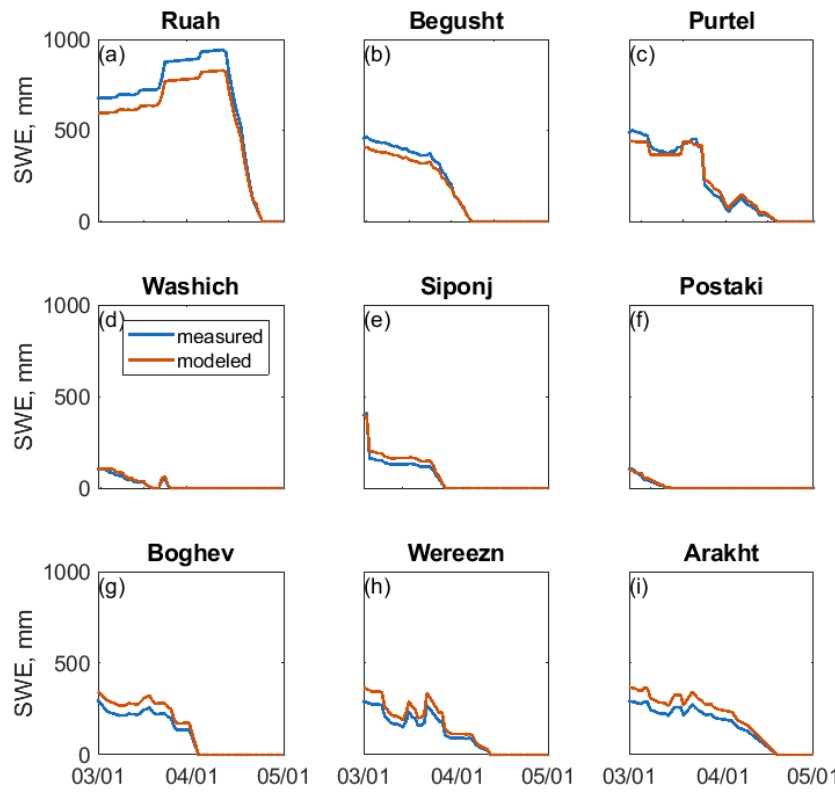

**Figure 6. Model vs. measured SWE at selected FOCUS stations for 2017 Mar to May.**

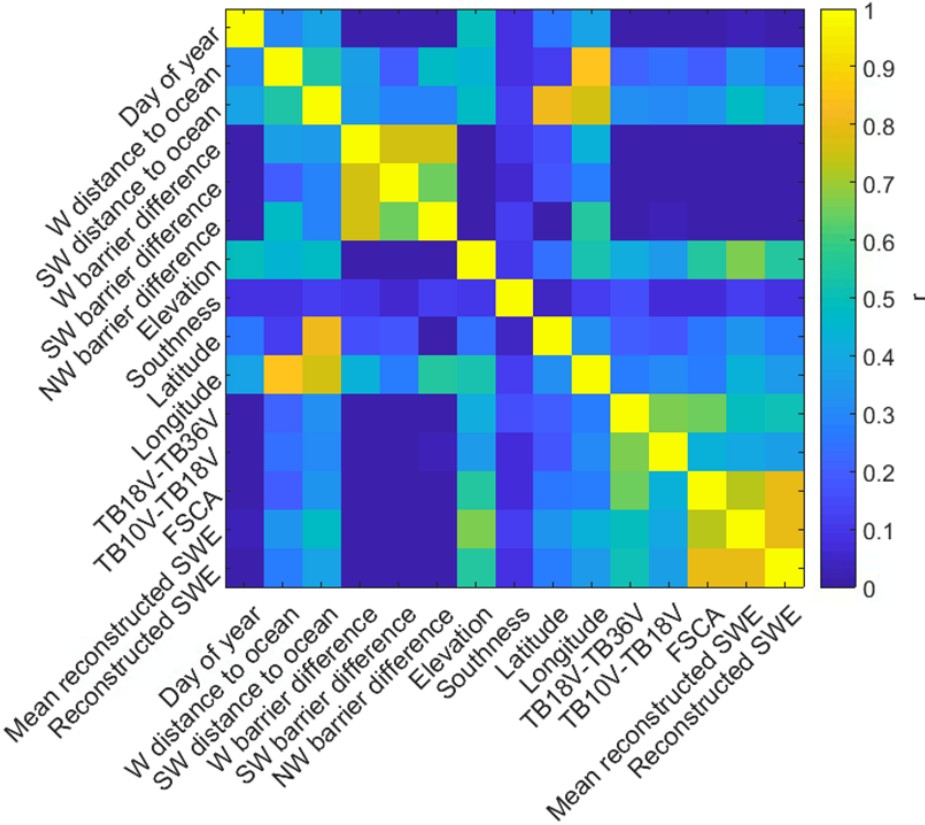

**Figure 7. Correlation coefficient (*r*) for all variables, sampled randomly across all years.**

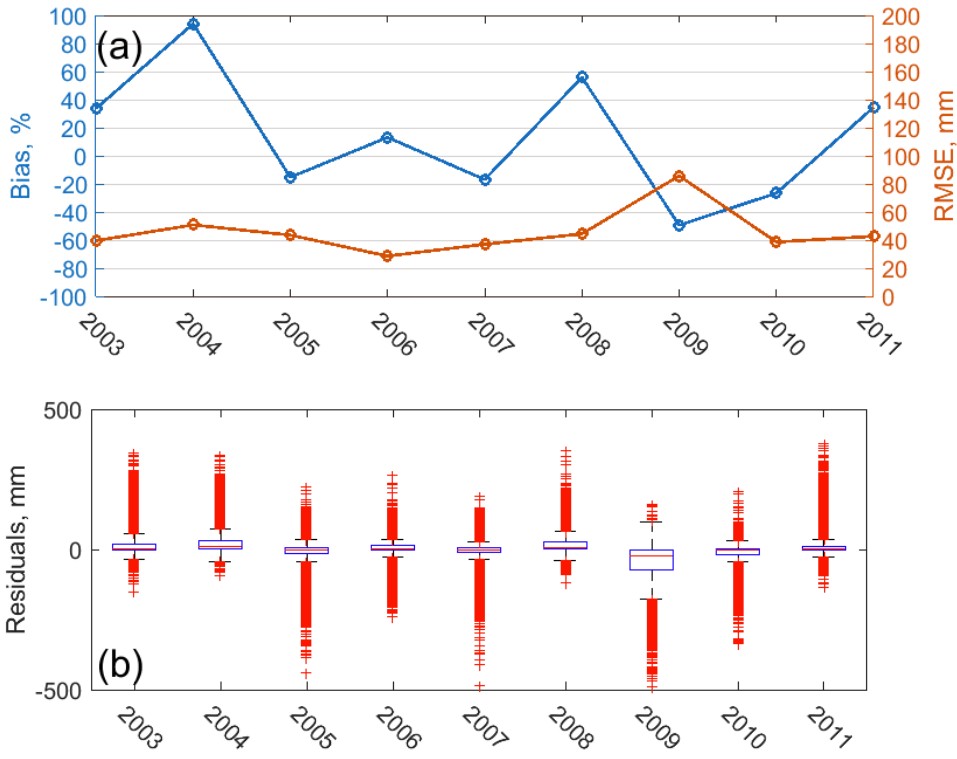

**Figure 8. (a) Bagged trees bias and RMSE; (b) residuals (machine learning prediction – reconstructed SWE) by year with red line as the median, boxes encompassing the 25th ($p_{25}$) and 75th ($p_{75}$) percentiles, whiskers showing non-outlier ranges, and crosses indicating outliers, defined as points greater/less than $p_{75} \pm w(p_{75} - p_{25})$, with $w = 1.5$.**

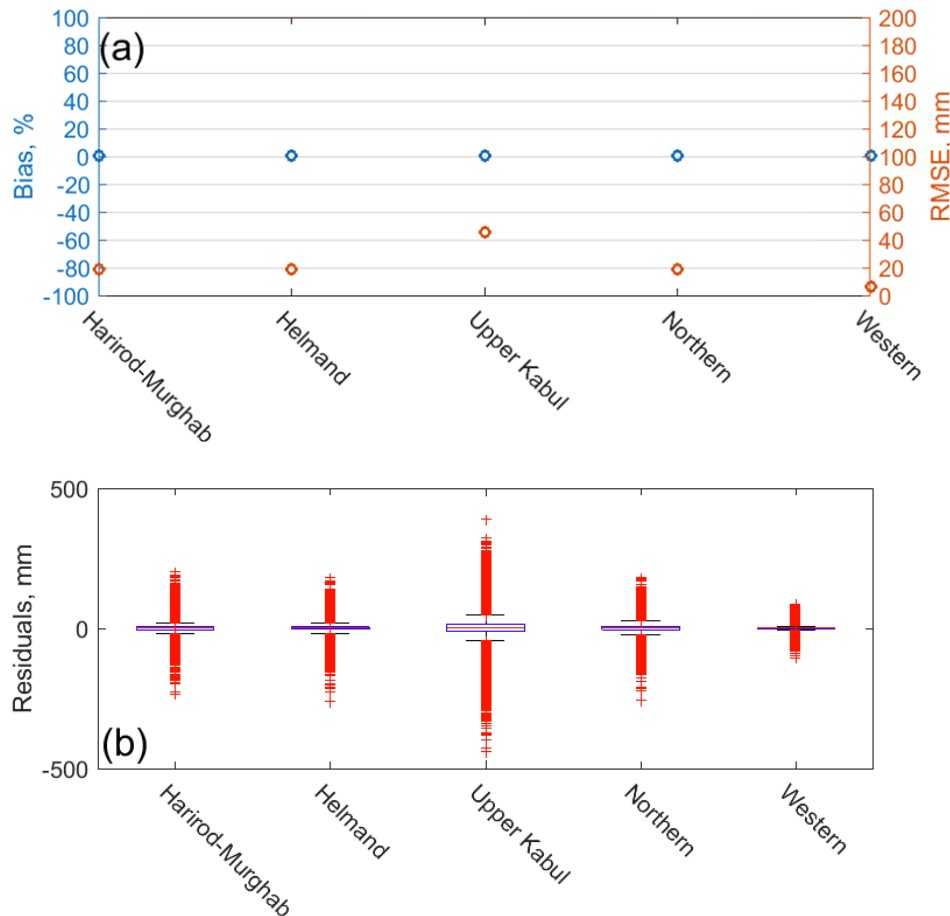

**Figure 9. Same as Figure 8 except by basin for all years, (a) Bagged trees bias and RMSE; and (b) residuals.**

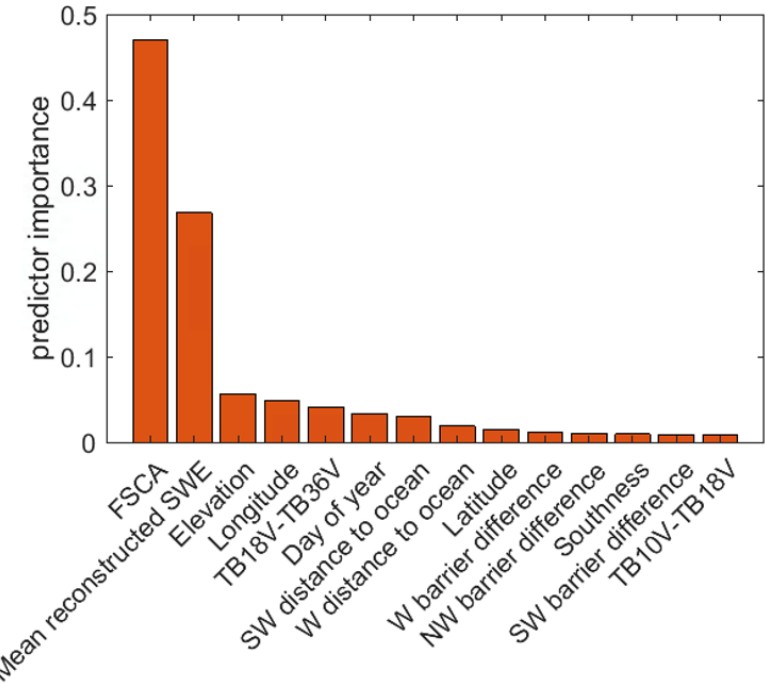

**Figure 10. Predictor importance (see Sect. 3.5.3) for randomly selected observations across all years using bagged regression trees.**

## 13      Tables

**Table 1. Climate records (monthly averages) from Kabul and Salang Pass.**

| Month | Mean air temp., °C | Mean precip., mm | Max snow depth, cm | Mean days with snow | Mean air temp., °C | Mean precip., mm | Max snow depth, cm | Mean days with snow |
|--------|------|------|------|------|------|------|------|------|
| | **Kabul, 1791 m (1956-1983)** | | | | **North Salang, 3366 m (1960-1983)** | | | |
| Jan | -2.3 | 34.3 | 48 | 7 | -10.3 | 108.7 | 300 | 13 |
| Feb | -0.7 | 60.1 | 65 | 6 | -9.5 | 142 | 367 | 15 |
| Mar | 6.3 | 67.9 | 30 | 3 | -5.4 | 185.9 | 383 | 18 |
| Apr | 12.8 | 71.9 | 0 | 0 | -0.1 | 197.8 | 450 | 15 |
| May | 17.3 | 23.4 | 0 | 0 | 2.9 | 123.7 | 366 | 10 |
| Jun | 22.8 | 1 | 0 | 0 | 7.5 | 10 | 66 | 1 |
| Jul | 25 | 6.2 | 0 | 0 | 9.7 | 6.8 | 0 | 0 |
| Aug | 24.1 | 1.6 | 0 | 0 | 8.6 | 6.7 | 2 | 0 |
| Sep | 19.7 | 1.7 | 0 | 0 | 4.7 | 7.5 | 40 | 1 |
| Oct | 13.1 | 3.7 | 0 | 0 | 0.7 | 30.2 | 102 | 6 |
| Nov | 5.9 | 18.6 | 4 | 0 | -4 | 68.4 | 184 | 8 |
| Dec | 0.6 | 21.6 | 30 | 4 | -7.8 | 104.3 | 320 | 11 |
| Annual | 12.05 | 312 | 14.75 | 2 | -0.25 | 992 | 215 | 8 |

**Table 2. Predictor and target variables**

| Name | Description |
|---|---|
| **Physiographic predictors** | |
| *Day of year* | Sine of day of year; sine function used to create a continuous variable |
| *Elevation* | Pixel average elevation |
| *Latitude* | Pixel center latitude |
| *Longitude* | Pixel center longitude |
| *NW/W/SW barrier difference* | Elevation difference between pixel and highest pixel in each direction, also called shield height |
| *W/SW distance to ocean* | Pixel distance to ocean or sea in each direction. NW was not used, as distance exceeds 5000 km for many pixels. |
| *Southness* | Computed as sin(slope)×cos(aspect), with slope as upward from horizontal and aspects from south with counter-clockwise as positive (Dozier and Frew, 1990) |
| **Dynamic predictors** | |
| $TB_{10V}–TB_{18V}$ | Difference between enhanced resolution PM brightness temperature at 10 GHz, vertically polarized, and 18 GHz, vertically polarized |
| $TB_{18V}–TB_{36V}$ | Difference between enhanced resolution PM brightness temperature at 18 GHz, vertically polarized, and 36 GHz, vertically polarized |
| $F_{SCA}$ | Fractional snow-covered area |
| *Mean reconstructed SWE* | Mean daily SWE computed over all years except the target year |
| **Target** | |
| *Reconstructed SWE* | Reconstructed daily SWE |

**Table 3. Basin area and April 1st reconstructed mean basin-wide SWE estimates from this study compared to passive microwave (AMSR-E) estimates from Daly et al. (2012). Highly glacierized basins are excluded.**

| | Years compared | Helmand | Western | Harirod-Murghab | Upper Kabul | Northern |
|---|---|---|---|---|---|---|
| Basin area, km$^2$ | - | 226,580 | 85,289 | 90,036 | 42,159 | 70,910 |
| SWE, mm, this study | 2003-2011 | 7 | 1 | 13 | 61 | 12 |
| SWE, mm, Daly et al. (2012) | 2002-2010 | 11 | 4 | 15 | 43 | 20 |

**Table 4. Error statistics for selected dates and different ways to account for uncertainty. See Section 3.4 for details.**

| | MAE, mm | MAE, % | Bias, mm | Bias, % |
|---|---|---|---|---|
| **Neighborhood and density uncertainty** | | | | |
| 1-Mar | 18 | 11% | -4 | -2% |
| 15-Mar | 16 | 14% | 5 | 4% |
| 1-Apr | 7 | 13% | -1 | -2% |
| **Neighborhood uncertainty only** | | | | |
| 1-Mar | 85 | 53% | -7 | -4% |
| 15-Mar | 81 | 70% | 21 | 18% |
| 1-Apr | 55 | 104% | -28 | -53% |
| **Density uncertainty only** | | | | |
| 1-Mar | 23 | 14% | -5 | -3% |
| 15-Mar | 18 | 16% | 3 | 3% |
| 1-Apr | 8 | 15% | -3 | -6% |
| **No uncertainty** | | | | |
| 1-Mar | 75 | 47% | -5 | -3% |
| 15-Mar | 88 | 77% | 47 | 41% |
| 1-Apr | 54 | 102% | 0 | 0% |