# Peer review of "Figure S1. Energy balance SWE reconstruction using the ParBal model with and without cold content compared to the mean of 2016 lysimeter outflow at the CRREL/UCSB Energy Balance Site (Bair et al., 2015)."

_The Cryosphere, 2017_

## Referee Comment (RC1) · J. Lundquist (Referee) · 15 Dec 2017

Overall, this is a very well written paper, and a solid contribution to the field. I would recommend it for publication following revisions.

There are two major issues that need strengthening before publishing.

1. The accuracy of the reconstruction method, which is treated as "truth" for the region's snowpack, needs to be more carefully assessed. In particular, all of the citations demonstrating the method works refer to the Sierra Nevada. Given what is known about the sources of weakness in the reconstruction method (e.g., cloud cover on the actual

date of disappearance, precipitation occurring after the date of peak SWE, errors in the atmospheric forcing terms), which issues are likely to be more or less problematic in Afghanistan compared to California and how are these errors likely to propagate to accuracy of the reconstructed SWE? (I realize that some issues may be hard to pin down, but the authors should be able to make some statements about the degree to which weather models are poorer in this region and the degree to which this will impact the reconstruction.) I would also like to see more analysis regarding potential issues with the daily-reset cold content of the snowpack in particularly cold regions. This could be examined by running a model in the traditional forward sense (with meteorological data from this region, fully accounting for multi-day accumulated "cold content") and comparing it to a model run in reconstruction mode.

2. The introduction nicely makes the connection that Afghanistan's water supply is susceptible to year-to-year variations in snowfall and that some way of making seasonal predictions of the snow available for runoff is very important. This paper demonstrates a way of doing this. However, the paper needs to clearly make the connection of how the errors inherent in the proposed method (order of 20%) compare to the errors in the current system. For example, what is the interannual variability in snowpack? How wrong would a water manager be if he/she just presumed mean runoff from snow? What methods are currently used for such a forecast, and what are their errors? (Are there any citations on this?) I'm guessing that 20% error is better than the current situation, but the actual numbers (or a best guess to the actual numbers) should be presented in the discussion and conclusions.

Some more minor issues include:

1. Given that only fSCA and mean reconstructed SWE had predictive power, why not test a simpler model with just those terms? How does that compare with the full set? Also, given the conclusion that only those variables mattered, why does the conclusion say that an operational system would need to ingest Passive Microwave data? Does that make a difference that warrants the effort?

2. The discussion should also address the implications of combined error from the forward prediction (which was trained on reconstructed SWE) and the errors in the reconstructed SWE (which the one point check suggests may be biased low 20%). How large might these combined errors be, and combined, are the expected errors still better than a baseline assumption of an average year?

Note, I am also providing an annotated manuscript to the editorial office and the authors, which marks in the text where the issues summarized here arise.

---

## Referee Comment (RC2) · S.R. Fassnacht (Referee) · 31 Dec 2017

I reviewed the previous version of this paper and my main issues were that the methods were not well described, and that the Results and Discussion section was very brief. The authors have addressed some of my comments, and the Methods section is much improved. However, the authors did not address some of the specifics comments that I had on the methods (see below). The Results and Discussion section is improved, but the authors need to further put the results into context. It is unclear what the error statistics (bias and RMSE) are based on - what is the "truth" that these statistics are being compared to.

[Figure]

While there is not much snow data available for Afghanistan, there is a USGS-USAID tool that may be relevant to this work: <https://earlywarning.usgs.gov/fews/software-tools/10>. There are specific results on SWE in Afghanistan presented by Daly et al. (2012); the authors should present more specific SWE results and reduce the emphasis on evaluation statistics (see specific comments below). They should consider more specifics than just the summary in Table 3.

Specific Comments

- Methods and Table 1: southness uses an aspect starting in the south. Previous papers have used northness, so at least a reference explaining the difference would be good.

- section 3.1: not sure how common these variables are. They are used for the Colorado River by Fassnacht et al. (2012).

- page 7, line 24 and Table 1: Barrier difference is also called shield height (e.g., Fassnacht et al., 2012).

- page 11, lines 10-11: Fassnacht et al. (2012) describes some of the physiographic predictor variables.

Overall the figures have been improved, but there is repetition in some of the Figure and Tables. - Figure 2b is a repeat of Figure 2a.

- Figure 4 is a repeat of the top of Table 4. Either remove Figure 4 or add the bottom of Table 4 to Figure 4 and remove Table 4. The same is true for Figure 5a and Table 5 - remove one of these.

- Table 3 and 5 could be combined. At minimum they should state the watersheds in the same order. It would also be helpful to state the area off each watershed so the SWE estimates i Table 3 can be taken in context.

Reference

Fassnacht, S.R., Dressler, K.A., Hultstrand, D.M., Bales, R.C., and Patterson, G.G.: Temporal Inconsistencies in Coarse-scale Snow Water Equivalent Patterns: Colorado River Basin Snow Telemetry-Topography Regressions, Pirineos, 167, 167-186, doi: 10.3989/Pirineos .2011.166008, 2012.
* * *

---

## Author Comment (AC1) · 24 Feb 2018

Overall, this is a very well written paper, and a solid contribution to the field. I would recommend it for publication following revisions.
There are two major issues that need strengthening before publishing.

The accuracy of the reconstruction method, which is treated as "truth" for the region's snowpack, needs to be more carefully assessed. In particular, all of the citations demonstrating the method works refer to the Sierra Nevada. Given what is known about the sources of weakness in the reconstruction method (e.g., cloud cover on the actual date of disappearance, precipitation occurring after the date of peak SWE, errors in the atmospheric forcing terms), which issues are likely to be more or less problematic in Afghanistan compared to California and how are these errors likely to propagate to accuracy of the reconstructed SWE? (I realize that some issues may be hard to pin down, but the authors should be able to make some statements about the degree to which weather models are poorer in this region and the degree to which this will impact the reconstruction.)

We thank Referee #1 for her comments. Some of the comments regarding the accuracy of reconstruction in Afghanistan and ground truth in general were shared by Referee #2. Frankly, similar comments have been made by previous reviewers of this study also, which has made it difficult to get the study through peer review. We believe we've finally added a validation source that will address some of these concerns.

At the end of 2017 Oct, we received a dataset of in situ snow depth and other meteorological measurements from Afghanistan, Tajikistan, and Pakistan that we had been waiting on for about a year. Please see the revised Section 3.4 on Validation in Afghanistan. Although these measurements are all manual and do not include SWE, only snow depth, they are far better than any other ground truth accessible to us, or likely available to anyone. For instance, through CRREL, we have access to US military weather station measurements in Afghanistan, which are sadly all going offline as they abandoned. Yet these weather stations provide no snow measurements. We have also never seen manual snow pit measurements from Afghanistan.

We find that using these in situ snow measurements, albeit with point to area extrapolation problems, geolocational uncertainty in MODIS pixel locations, and uncertainty in the density model, provide a better picture of the accuracy of our SWE reconstructions than speculation about cloud cover, quality of forcings, and precipitation after peak SWE.

I would also like to see more analysis regarding potential issues with the daily-reset cold content of the snowpack in particularly cold regions. This could be examined by running a model in the traditional forward sense (with meteorological data from this region, fully accounting for multi-day accumulated "cold content") and comparing it to a model run in reconstruction mode.

Likewise, we find that a full evaluation of the daily cold content scheme not possible with the data available to us in Afghanistan. To fully evaluate whether or not the pack is ripe, snow pit, bulk temperature, or lysimeter measurements are needed. At the least, to run a model like SNOWPACK, hourly energy and mass balance forcings with snow depth or SWE are needed. We still do not have in situ measurements of these forcings available anywhere in the watersheds of Afghanistan.

In our view, the cold content scheme has been shown to work well in the Sierra Nevada and Rocky Mountains (Jepsen et al) and at predicting the onset of lysimeter discharge (this study). We now know from the FOCUS station measurements that air temperatures are quite warm, which is why all the stations classified as warm snow types: alpine, maritime, or prairie, We

conclude that most areas of Afghanistan have a snow climate similar to the Rockies (alpine) and the Sierra (maritime). Thus, we suggest that the cold content scheme can be justifiably applied to most of Afghanistan's watersheds. We agree that the daily cold content scheme should be tested in cold regions (e.g. taiga and tundra snowpacks). There are probably areas like this in Afghanistan above 6000 m, but do not have the in situ measurements to test this claim. In fact, we are not aware of any areas in all of High Mountain Asia with full energy/mass balance instrumentation at these altitudes.

2. The introduction nicely makes the connection that Afghanistan's water supply is susceptible to year-to-year variations in snowfall and that some way of making seasonal predictions of the snow available for runoff is very important. This paper demonstrates a way of doing this. However, the paper needs to clearly make the connection of how the errors inherent in the proposed method (order of 20%) compare to the errors in the current system. For example, what is the interannual variability in snowpack? How wrong would a water manager be if he/she just presumed mean runoff from snow? What methods are currently used for such a forecast, and what are their errors? (Are there any citations on this?) I'm guessing that 20% error is better than the current situation, but the actual numbers (or a best guess to the actual numbers) should be presented in the discussion and conclusions.

This is a good point and we agree that our errors should be put into context in terms of operational utility. There is little in the way of water management in Afghanistan so we'll have to use hypothetical examples. More common is that snow and glacier melt fed streams run dry in the fall without warning (e.g. Introduction).

Please see our addition of Nash-Sutcliffe efficiencies on p 10 l 1-7, which show improvement over a mean forecast for all years.

Note that Figure 2b already shows the interannual variability in absolute and relative terms. We've added a few sentences describing these figures (p 5 l 28-31)

Some more minor issues include:
1. Given that only fSCA and mean reconstructed SWE had predictive power, why not test a simpler model with just those terms? How does that compare with the full set?

A model with just those predictors should work almost as well. Building such a model is certainly less effort and something we will try in the future based on our results. However, we fail to see why a simpler model needs to be employed here given that the machine learning techniques employed should be very robust against overfitting. Moreover, the passive microwave data, while not correctly estimating deep snow, help with the shallow snow and in areas with more cloud cover could conceivably help with estimating snow-covered area.

Bagged trees and subsequently random forests were invented to prevent overfitting. In our case, the hyperparameters for the bagged trees–which include the number of variables sampled for each tree, the minimum leaf size, and the maximum number of split–were optimized by minimizing a cross-fold validation error. The end result should be that unimportant variables do not affect the model.

The neural network model is more of a black box with more potential for overfitting, however since the results are nearly identical to the bagged trees, we conclude that overfitting is not an issue.

Also, given the conclusion that only those variables mattered, why does the conclusion say that an operational system would need to ingest Passive Microwave data? Does that make a difference that warrants the effort?

Ok, we've removed the PM reference part from the conclusion.

2. The discussion should also address the implications of combined error from the forward prediction (which was trained on reconstructed SWE) and the errors in the reconstructed SWE (which the one point check suggests may be biased low 20%).

With added section 3.4, we find it unlikely that our reconstructed SWE results are biased and given the uncertainty in our density model, they have very low errors.

How large might these combined errors be, and combined, are the expected errors still better than a baseline assumption of an average year?

We've added NS statistics from Section 4 to the conclusion.

Note, I am also providing an annotated manuscript to the editorial office and the authors, which marks in the text where the issues summarized here arise.

Needhamn, J.: Water Balance and Regulation Alternative Analysis for Kajakai Reservoir using HEC-ResSim, US Army Corps of Engineers, 58, 2007.
Vuyovich, C., and Jacobs, J. M.: Snowpack and runoff generation using AMSR-E passive microwave observations in the Upper Helmand Watershed, Afghanistan, Remote Sensing of Environment, 115, 3313-3321, doi 10.1016/j.rse.2011.07.014, 2011.

---

## Author Comment (AC2) · 24 Feb 2018

I reviewed the previous version of this paper and my main issues were that the methods were not well described, and that the Results and Discussion section was very brief. The authors have addressed some of my comments, and the Methods section is much improved. However, the authors did not address some of the specifics comments that I had on the methods (see below). The Results and Discussion section is improved, but the authors need to further put the results into context. It is unclear what the error statistics (bias and RMSE) are based on - what is the "truth" that these statistics are being compared to.

We thank Referee #2 for the critiques and for reviewing the paper a second time. We understand how much time these reviews take and are grateful for the feedback.

As for explaining what the error statistics are based on and what the "truth" is, we state on p 9 l 11-12 that 20% of the observations were held out for validation. Also, please see added Section 3.4 on in situ validation with snow depth from FOCUS weather monitoring posts in Afghanistan and our responses to Referee #1 concerning validation of our reconstructed SWE estimates.

Overall, we find that our reconstructed SWE estimates have proven to be unbiased and accurate across a variety of snow climates. The addition of the FOCUS snow depth observations as validation source further strengthens this statement. Therefore, we suggest our reconstructed SWE estimates represent the most accurate "truth" available for Afghanistan.

While there is not much snow data available for Afghanistan, there is a USGS-USAID tool that may be relevant to this work: <https://earlywarning.usgs.gov/fews/software- tools/10>. There are specific results on SWE in Afghanistan presented by Daly et al. (2012); the authors should present more specific SWE results and reduce the emphasis on evaluation statistics (see specific comments below). They should consider more specifics than just the summary in Table 3.

We are very familiar with the FEWS NET product as it is based on passive microwave estimates of the snowpack in Afghanistan. These assessments are done by CRREL on a weekly basis for operational use. The approach used is summarized in two previous studies, both of which are cited in the manuscript (Daly et al., 2012; Vuyovich and Jacobs, 2011). In fact, the original funding for this project was to use our reconstructed SWE estimates to improve these weekly snow assessments. In Table 3, we specifically compare our reconstructed SWE estimates to those from Daly et al. (2012). The basin wide SWE estimates appear close but that is somewhat misleading as large areas of Afghanistan's watersheds never have snow cover. We suggest that it is more important to concentrate on the heavier snow areas in each basin, all of which have pixels with reconstructed SWE values an order of magnitude greater than the passive microwave saturation limit of 150 mm (p5 l34-35;6 l1-2).

We have found that the passive microwave estimates are not only inaccurate in terms of predicting snow depth or SWE, but they fail to even capture the rank order statistics (e.g. Fig S3), which are arguably more important to a water manager than "million cubic meters" of SWE. This is one reason why it is not an important predictor, even at the enhanced 3.125 km resolution.

Specific Comments
- Methods and Table 1: southness uses an aspect starting in the south. Previous papers have used northness, so at least a reference explaining the difference would be good.

Ok, Dozier and Frew (1990) citation added to Table (now) 2. The convention referencing directions to 0 degrees at south dates back to Sellers' *Physical Climatology* and Geiger's *Climate Near the Ground*. Moreover, such usage is consistent with a right-hand coordinate system (which measuring directions clockwise is not).

- section 3.1: not sure how common these variables are. They are used for the Colorado River by Fassnacht et al. (2012).

Ok reference added, p3 l 7.

- page 7, line 24 and Table 1: Barrier difference is also called shield height (e.g., Fassnacht et al., 2012).

We are unclear about this comment. The entry in Table 2 says "also called shield height".

- page 11, lines 10-11: Fassnacht et al. (2012) describes some of the physiographic predictor variables.

Ok, reference added

Overall the figures have been improved, but there is repetition in some of the Figure and Tables.

- Figure 2b is a repeat of Figure 2a.

Referee #1 specifically requested that we show and discuss inter-annual variability with respect to the mean so we've kept Figure 2b.

- Figure 4 is a repeat of the top of Table 4. Either remove Figure 4 or add the bottom of Table 4 to Figure 4 and remove Table 4. The same is true for Figure 5a and Table 5 - remove one of these.

Ok, tables (now) 5&6 have been deleted.

- Table 3 and 5 could be combined. At minimum they should state the watersheds in the same order. It would also be helpful to state the area off each watershed so the SWE estimates i Table 3 can be taken in context.

See last response and we've added the basin area as a row in (now) Table 3.

Reference

Fassnacht, S.R., Dressler, K.A., Hultstrand, D.M., Bales, R.C., and Patterson, G.G.: Temporal Inconsistencies in Coarse-scale Snow Water Equivalent Patterns: Colorado River Basin Snow Telemetry-Topography Regressions, Pirineos, 167, 167-186, doi: 10.3989/Pirineos .2011.166008, 2012.

Daly, S. F., Vuyovich, C. M., Deeb, E. J., Newman, S. D., Baldwin, T. B., and Gagnon, J. J.: Assessment of the snow conditions in the major watersheds of Afghanistan using multispectral and passive microwave remote sensing, Hydrological Processes, 26, 2631-2642, doi 10.1002/hyp.9367, 2012.

Dozier, J., and Frew, J.: Rapid calculation of terrain parameters for radiation modeling from digital elevation data, IEEE Transactions on Geoscience and Remote Sensing, 28, 963-969, doi 10.1109/36.58986, 1990.

Vuyovich, C., and Jacobs, J. M.: Snowpack and runoff generation using AMSR-E passive microwave observations in the Upper Helmand Watershed, Afghanistan, Remote Sensing of Environment, 115, 3313-3321, doi 10.1016/j.rse.2011.07.014, 2011.

---

## Author Response (AR2)

Dear authors,

Thanks for your submission of the revised manuscript.
Below is a list of minor/technical revision that have been reported by the last referee :

- p.6-7: Is it possible to add a table indicating the criteria used for the snow climate classification, for each class. It would also be possible to merge this table with Fig. 4, which does not give so much information. From p.7, l.8-10, I understand that a snow density model is defined for each snow climate class, but I might be wrong. It is possible to explain more clearly why you need this snow climate classification before (at p.6)?

- p.7: I did not quite understand what you mean by "density uncertainty", do you obtain different scenarios from the the snow density model?
'
Can you please take them into account ?

Thanks a lot,

Best regards,

Marie Dumont
* * *
Dear Prof. Dumont:

In response to the third reviewer's comments, we have added the following lines.

Per the 1st comments, we have added:

p. 6 l 31 "and associated density model"

[revised manuscript text omitted]